Exploring quadriceps activity variations in women: the role of seated posture during stair climbing

Chun SaBin 1
Park JongChul 2 3 jcpark@pknu.ac.kr
1 Department of Physical Education, Pukyong National University , Busan , Republic of South Korea
2 Major of Marine Sports, Division of Smart Healthcare, Pukyong National University , Busan , Republic of South Korea
3 Marine Designeering Education Research Group, Pukyong National University , Busan , Republic of South Korea
Espada Mário
Electronic publication date: 2024 Nov 22
Publication date: 2024
Volume: 12
Electronic Location ID: e18547
Received 2024 May 15; Accepted 2024 Oct 28
Copyright: © 2024 Chun and Park
Copyright year: 2024
Copyright holder: Chun and Park
License: This is an open access article distributed under the terms of the Creative Commons Attribution License, which permits unrestricted use, distribution, reproduction and adaptation in any medium and for any purpose provided that it is properly attributed. For attribution, the original author(s), title, publication source (PeerJ) and either DOI or URL of the article must be cited.
License URL: https://creativecommons.org/licenses/by/4.0/

Keywords: Quadriceps muscle, Muscle activity, Women, Sitting position, Stair walking

Funding: The authors received no funding for this work.

==============================
Background

Adults in modern society spend most of their time in a sitting position. However, sitting for long periods of time can affect the length and sensation of the quadriceps muscle and negatively affect the performance of functional tasks required in daily life. In addition, it may have different characteristics according to sitting postures.

Aim

The aim of this study is to investigate the differences in quadriceps muscle activity during stair climbing and descending based on various sitting postures.

Methods

Thirty healthy adult women were classified into sitting upright (SU), sitting with legs crossed (SLC), and sitting with ankles crossed over knees (SAC). The muscle activities of the vastus medialis (VM), vastus lateral (VL), and rectus femoris (RF) were collected during climbing and descending the stairs three times each. After which a sitting posture was maintained for 15 min, the post-measurement was performed in the same way as the pre-measurement. One-way analysis of variance or Kruskal-Walli’s test was used to check homogeneity between each group, least significant difference (LSD) was used for post-hoc testing. Paired-sample t-test or Wilcoxon signed-rank test was used to check before-and-after differences within the group. Repeated measures analysis of variance was used to determine between-group and within-group interactions.

Results

VM muscles were significantly decreased after maintaining a sitting posture during descending stairs of SAC group (p = 0.047). In conclusion, this study indicates that maintaining a SAC sitting posture for a certain period causes a temporary decrease in muscle activity of the VM muscle during descending stairs. This provides important implications for early treatment to manage knee pain and strengthen quadriceps muscles in women. Future studies should focus on groups with patellofemoral pain syndrome or knee pain to better understand the impact of sitting posture on quadriceps activity.

Introduction

Sitting is an essential activity in daily life, such as watching TV, playing video games, working, and relaxing. However, sitting life corresponds to activities with a metabolic equivalent task (MET) of less than 1.5 MET, such as lying down and sleeping, which do not increase energy expenditure (Pate, O’neill & Lobelo, 2008), maintaining a sedentary lifestyle for extended periods has been reported to have negative effects on health and longevity (Pate, O’neill & Lobelo, 2008; Van der Ploeg et al., 2012; Ozemek & Arena, 2021). Studies have confirmed that modern adults spend up to 9 h of their waking hours sitting (Thorp et al., 2010), it has been reported that sitting for a long time increases hip joint pressure and causes muscle stiffness and fatigue, which can affect the performance of functional tasks required in daily life (Kett & Sichting, 2020; Lee, Park & Yoo, 2011). Not only that, evidence has been reported that even temporary stretching or maintaining a static posture for as little as 15 min can have negative effects on the functional activity of the thigh muscles in the lower extremities, including delays and changes in muscle activity (Irawan, Sinsurin & Sonsukong, 2022; Sánchez-Zuriaga, Adams & Dolan, 2010).

The quadriceps muscle consists of the vastus medialis, vastus lateral, and rectus femoris muscles, surrounds the knee joint from the front of the thigh and is involved in knee extension and flexion. In additionally, the patella is in the pulley groove at the bottom of the femur and serves as a lever during knee extension and provides a mechanical advantage to the quadriceps. Normal tracing of the patella depends on the structure of the kneecap, but also on the magnitude and direction of forces in the soft tissue structures surrounding the knee joint (Han, Sawatsky & Herzog, 2022), it is reported that the vastus medialis is particularly affected (Han, Sawatsky & Herzog, 2022). Therefore, stability and normal tracking of the patella through the femoral muscles play an important role in functional movements required in daily life, such as walking, running, and going up and down stairs (Elias & White, 2004). However, muscle weakness, changes in line of action, and muscle imbalances can cause the patella to deviate from its normal tracking, causing knee pain (Lorenz et al., 2012). Even, anterior knee pain can occur due to a variety of potential reasons, such as increased patella pressure, tibial rotation, and impaired normal tracking (Cesarelli, Bifulco & Bracale, 1999; Lorenz et al., 2012).

Although the benefits of physical activity are well known, the adverse effects of prolonged sitting are a relatively new topic of interest that has not yet been extensively studied. Because most studies looked at lower extremity kinematics and muscle activity while seated on the floor, the results may not be representative of non-floor sitters or common postures. Therefore, maintaining a sitting position on a chair for a long period of time and investigating differences in quadriceps muscle activity according to posture characteristics can provide understanding of how the sitting posture used in a large population affects the quadriceps femoris neuromuscular control mechanism. Additionally, given the more frequently reported knee pain in women (Fulkerson & Arendt, 2000; Glaviano, Boling & Fraser, 2021; Culvenor et al., 2019), and the functional aspects of the quadriceps femoris, understanding changes in quadriceps activity immediately after maintaining a static posture may contribute to understanding the triggers of knee pain in women.

Therefore, in this study, we aimed to determine the differences in quadriceps muscle activity among healthy women when maintaining various sitting postures. Building on previous studies (Jung, Jung & In, 2020; Kang et al., 2012; Snijders et al., 1995), we focused on functional tasks performed before and after adopting three postures: sitting upright (SU), sitting with legs crossed (SLC), and sitting with ankles crossed on knees (SAC). The functional tasks were the ascending and descending stairs. The results of this study can provide improved knowledge for understanding the effects of sitting posture characteristics when maintaining a sitting posture for a long time and preventing and managing potential knee pain in woman.

Materials and Methods

Participants

In this study, 30 healthy women in their 20 s with left and right leg length <2 cm and Quadriceps-angle <20 degrees were recruited (Irawan, Sinsurin & Sonsukong, 2022), and all subjects completed the International Physical Activity Questionnaire (IPAQ). Based on their physical activity levels, the participants were classified into three groups using the Stratified Random Sampling method: SU group (n = 10), SLC (n = 10), and SAC group (n = 10). The sample size of 10 subjects per group was calculated using G*power (v3.1.9.2, Heinrich-Heine-Universität, DEU) software to detect a medium effect size (d = 0.6) with 88% power and a 5% significance level. The study was approved by the bioethics committee of the Pukyong National University (1041386-202211-HR-76-01), and all participants voluntarily consented after receiving a comprehensive explanation of the study’s objectives and procedures, and written informed consent was obtained each participant. The dominant foot was determined based on the leg used when kicking a ball (Behm & Chaouachi, 2011). Demographic characteristics of each group are presented in Table 1.

Table 1 Demographic characteristics of participants.

	Dominant leg (n, %)	IPAQ (n, %)	IPAQ (MET/min)	Height (cm)	Weight (kg)	Age (years)	
Right	Left	High	Moderate	Low	Mean ± SD	Mean ± SD	Mean ± SD	Mean ± SD	
SU (n = 10)	10 (100)	0 (0)	1 (10)	6 (60)	3 (30)	2,935.40 ± 2,922.44	164.60 ± 6.15	57.90 ± 6.96	21.30 ± 1.25	
SLC (n = 10)	10 (100)	0 (0)	0 (0)	7 (70)	3 (30)	2,837.70 ± 2,227.61	164.80 ± 2.65	57.50 ± 9.45	22.80 ± 3.32	
SAC (n = 10)	9 (90)	1 (10)	2 (0)	6 (60)	2 (20)	2,769.00 ± 2,812.74	160.70 ± 5.03	53.30 ± 6.94	21.90 ± 1.52	
t(p)/Z(p)	2.069 (0.355)	2.355 (0.671)	0.010 (0.990)	2.282 (0.121)	3.232 (0.199)*	0.836 (0.658)*	
Notes:

* Non-parametric statistics.

IPAQ, international physical activity questionnaire; SU, sitting upright; SLC, sitting with legs crossed; SAC, sitting with ankles crossed over knees group.

Procedures

All subjects performed the functional task of going up and down the stairs three times each at a self-selected speed on the emergency stairs in the building (high 16 cm, width 29 cm, marble material; Fig. 1). All subjects performed a task of going up and down a single staircase with one leg on a set of 12 stairs. A 30 second (s) rest was provided after climbing stairs and a 60 s rest after climbing stairs. If the subject unconsciously grabbed the wall or handrail, or if there was a malfunction of the equipment, the data was immediately discarded and remeasured. After the pre-measurement, subjects were requested to sit for 15 min in a posture as classified, using a chair with a height of 46.5 cm and without a backrest. The SU kept the ankles and torso vertical from the ground, and SLC kept the body vertical from the ground, bringing the knees together and crossing the legs. The SAC kept the torso perpendicular to the ground, knees apart and one ankle resting on the other knee (Fig. 2). Additionally, personal smartphones were used to ensure that the body was not unconsciously supported by a desk or other objects. Once the time limit was over, climbing and descending the stairs were performed three times each, following the same procedure as in the pre-measurement (Irawan, Sinsurin & Sonsukong, 2022).

Figure 1 The stairs used when going up and down stairs.

The square box at the top right indicates the staircase and height.

Figure 2 Sitting position.

Sitting upright group (A), sitting with legs crossed group (B), and sitting with ankles crossed over knees group (C).

Electromyography

All participants wore shorts for quadriceps muscle activity measurement. The muscle activities of the vastus medialis (VM), vastus lateralis (VL), and rectus femoris (RF) were measured using a wireless surface electromyography (EMG) Noraxon mini DTS sensors (Noraxon, Scottsdale, AZ, USA) at a sampling rate of 1,500 Hz. The disposable surface electrode (single electrode T246H) made of Ag/Agcl was attached to the muscle abdomen by referring to Surface Electromyography for the non-invasive Assessment of Muscles (SENIAM, n.d.; Fig. 3). To minimize resistance, the skin surface of each muscle was prepared by removing hair and cleaning with alcohol cotton, and the electrodes were then affixed by two researchers. The maximum voluntary isometric contraction (MVIC) of each muscle was measured using a leg-extension machine. All subjects were instructed to hold the handle and extend their knees with maximum effort while keeping their hips and knees flexed at 90°, and to hold this position for 5 s (Amiri-Khorasani & Kellis, 2013). Sufficient rest was provided between three repetitions.

Figure 3 Comparison of the vastus medialis activity during descending stairs (expressed in %MVIC).

SU, sitting upright; SLC, sitting with legs crossed; SAC, sitting with ankles crossed over knees group.

All data were collected using Noraxon MR3 3.14 software. The raw data of each muscle were processed by applying a band-pass filter of 20 to 450 Hz to remove noise, and a root mean square (RMS) smoothing with a window of 100 milliseconds (ms) was applied. The analysis time point was selected as 50 ms before and after the maximum value of each movement during climbing and descending stairs, which is a functional performance task. An average value of 100 ms in total was calculated for each time point (Irawan, Sinsurin & Sonsukong, 2022). The quadriceps activity during both stair climbing and down was normalized by calculating the percentage of MVIC (%MVIC) (Mok, Bahr & Krosshaug, 2018). Values exceeding MVIC were treated as missing values and used in analysis.

Statistical analysis

All analyses were performed using IBM SPSS version 27.0 (Armonk, NY, USA). Normal distribution was evaluated using the Shapiro–Wilk test. One-way analysis of variance (ANOVA) or Kruskal-Walli’s test were used to identify demographic differences and differences in quadriceps muscle activity according to sitting posture, a paired t-test or Wilcoxon signed-rank test were conducted to confirm the difference between the groups before and after maintaining the sitting posture. Least significant difference (LSD) was used for post-hoc testing. Repeated-measures analysis of variance was used to confirm the interaction effect between sitting posture and sitting posture maintenance. All results are presented as mean ± standard deviation (SD), and the statistical significance level was set at p < 0.05.

Results

Table 2 shows the results for the quadriceps muscle activity during stairs walking according to sitting postures and maintained sitting postures. Aa a result of confirming quadriceps activity during stair climbing and descending before and after applying three different postures for a long time, there was no difference in quadriceps activity during stair walking before and after maintaining all sitting posture. However, in the SAC group while stair descending, the VM muscle activity was significantly lower after (35.00 ± 9.98 %MVIC) maintaining the sitting posture compared to before (51.82 ± 18.84 %MVIC) maintaining the sitting posture (t = 2.341, p = 0.047; Fig. 4). The interaction effect maintaining a sitting posture for a long time and sitting posture was not confirmed for all muscles during stair climbing and descending.

Table 2 Difference in quadriceps muscle activity (expressed in %MVIC) between and within pre-and post-intervention during stairs walking depending on sitting posture.

	Muscle	Group	Pre	Post	t(p)	F(p)	
Mean ± SD	Mean ± SD	
Stair climbing	VM	SU	71.99 ± 19.89	64.87 ± 15.71	2.010 (0.091)	–	
SLC	71.40 ± 20.04	69.26 ± 20.12	0.618 (0.554)	
SAC	66.31 ± 18.11	61.01 ± 21.01	1.894 (0.091)	
t(p)	0.238 (0.790)	0.438 (0.650)		
VL	SU	73.07 ± 19.44	69.14 ± 16.00	0.463 (0.659)	–	
SLC	62.96 ± 21.54	62.42 ± 18.01	0.121 (0.907)	
SAC	72.82 ± 10.54	78.98 ± 13.08	−1.841 (0.108)	
t(p)	0.885 (0.427)	2.300 (0.124)		
RF	SU	40.77 ± 14.58	44.30 ± 16.61	−0.948 (0.371)	–	
SLC	46.07 ± 25.00	41.61 ± 25.28	−1.256 (0.256)	
SAC	39.13 ± 15.31	44.86 ± 20.61	−1.638 (0.136)	
t(p)	0.347 (0.710)	0.055 (0.947)		
Stair descending	VM	SU	57.42 ± 21.94	42.67 ± 16.31	2.320 (0.053)	3.279 (0.055)	
SLC	47.80 ± 15.23	50.03 ± 18.33	−0.562 (0.588)	
SAC	51.82 ± 18.84	35.00 ± 9.98	2.341 (0.047)	
t(p)	0.595 (0.559)	2.240 (0.127)		
VL	SU	50.83 ± 17.58	43.69 ± 14.49	2.119 (0.063)	–	
SLC	36.02 ± 10.82	44.73 ± 22.70	−0.760 (0.469)	
SAC	46.43 ±7.59	45.03 ± 6.74	0.771 (0.460)	
t(p)	3.322 (0.052)	0.019 (0.981)		
RF	SU	34.30 ± 17.90	32.33 ± 10.19	0.544 (0.599)	–	
LC	36.94 ± 29.18	38.17 ± 31.39	−0.051 (0.959)*	
SAC	31.61 ± 14.41	33.39 ± 18.25	−0.589 (0.570)	
t(p)	0.155 (0.857)	0.209 (0.901)*		
Notes:

* Non-parametric statistics.

VM, vastus medialis; VL, vastus lateralis; RF, rectus femoris; SU, sitting upright; SLC, sitting with legs crossed; SAC, sitting with ankles crossed over knees group.

Figure 4 Quadriceps electrode attachment site.

Discussion

This study was conducted to determine the effects of maintaining a sitting position for long periods of time and sitting posture characteristics on the quadriceps muscles. The main finding of this study was that the activity of the VM muscle decreased after 15 min of sitting in the SAC groups during stair descending

These results thought to be related with previous research (Beutler et al., 2002; Sousa & Tavares, 2012) that has demonstrated a decrease in muscle strength and activity in the quadriceps muscle immediately after static stretching. Prolonged sitting posture may lead to changes in neural factors, such as alterations in golgi tendon organ reflex activity, mechanoreceptors, and pain feedback, or fatigue-related mechanisms, which could contribute to changes in quadriceps muscle activity and spasticity (Behm & Chaouachi, 2011; Morse et al., 2008; Winchester, Nelson & Kokkonen, 2009). In addition, static stretching is known to reduce force generation ability by increasing the compliance of the muscle-tendon units, and it has been reported that muscle elasticity characteristics can change when the muscle is held in a stretched position for a prolonged period (Amiri-Khorasani & Kellis, 2013). Therefore, these results suggest that the effect of muscle mechanical elasticity immediately after maintaining a sitting posture may be minimal.

Other results observed in muscle groups performing the same function suggest that the length of the vastus medialis tendon or muscle may be influenced by strain and neural factors when considering the postural characteristics of SAC, such as external rotation of the femur, flexion of the hip joint, and adduction of the tibia by placing the ankle on the knee. Additionally, although the quadriceps muscles contribute to both extension and flexion of the knee, the vastus medialis is specialized for the movements necessary to stabilize the patella, while the vastus lateralis, in contrast, can maintain a constant level of activity to support the quadriceps in a variety of static postures (Silva et al., 2014). Therefore, we believe that significant changes were found only in the vastus medialis muscle. However, the reduced action of the vastus medialis, which has a horizontal direction of contraction compared to other muscles, is explained as an important factor in patella kinematics (Lin et al., 2010; Powers, 2000). Therefore, in activities requiring knee flexion, the decrease in medial force may create a dynamic relationship that increases lateral force, potentially exacerbating pain caused by the imbalance of the quadriceps muscles. In fact, it has explained that VM muscle weakness and delayed activity can cause patella lateral displacement and cause knee joint diseases such as patellofemoral pain syndrome in woman (Arumugam & Parikh, 2015; Grabiner, Koh & Draganich, 1994). Thus, in these results could potentially be important for the pathogenesis, progression, and management of individuals with knee osteoarthritis (Hinman et al., 2002). On the other hand, previous studies have not explained the relationship between people who have a habit of sitting with their knees bent and holding them for a long time with knee pain (Arumugam & Parikh, 2015). Also, since muscle activity cannot be directly correlated with knee pain, these limitations must be considered when interpreting the results.

The commonly reported meaning of correct posture or good posture means that it contains a minimum of stress and strain and is conducive to maximum efficiency of body use (Claus et al., 2009). However, because of observing the characteristics of sitting posture with SU, SLC, and SAC group in this study, muscle activity of the vastus medialis muscle decreased during descending stairs in the group with SAC group. However, since this study only observed muscle activity in the quadriceps muscle while performing functional tasks after maintaining a sitting posture for a long time, there may be insufficient evidence that these changes immediately affect the efficiency of body use. Also, this study did not quantify the pressure or force exerted on the joints due to posture, the direct impact of posture on the joints cannot be considered. However, evidence suggests that prolonged sitting elevates hip pressure and accelerates fatigue (Lee, Park & Yoo, 2011; Irawan, Sinsurin & Sonsukong, 2022). Considering the structural aspects of the SAC posture, it appears that significant pressure will be placed on the knee joints and hip joints. Therefore, the results of this study provide a basis for rejecting the SAC posture.

Although no change in quadriceps muscle activation was observed during stair climbing in this study, the decrease in muscle activation of the vastus medialis muscle confirmed during stair descent is thought to be related to differences in gait characteristics. Climbing stairs during stair walking consumes a lot of energy in the knee joint to lift and move the body vertically and horizontally and is characterized by requiring high joint loads and moments (McFadyen & Winter, 1988). On the other hand, descending stairs has the characteristics of short single limb support time, anteroposterior ground reaction force directed backward, and using the ankle joint to reduce the flexion moment of the knee joint (Liikavainio et al., 2007; Spanjaard et al., 2008). In fact, in a previous study conducted to investigate differences in gait patterns, time, and muscle activity of the lower extremities when going up and down stairs, it was confirmed that different gait patterns were shown when going up and down stairs (Kim, Kim & Seo, 2006; Nam & Lee, 2017). However, the maximum vertical force confirms when descending the greater than ascending the stairs. One of the key mechanisms for protecting the lower limbs and the whole body from impact loading is the pre-activation of muscles before ground contact, which has been described as important in controlling shock loading, particularly in individuals with asymptomatic knee osteoarthritis (Liikavainio et al., 2007). While this study recruited healthy subjects and muscle activities were used in the analysis, however, prolonged postures or stretched muscles, it suggests that may lead to alterations in the musculoskeletal protective mechanisms against impact loading such as during stair descent.

The quadriceps muscle plays a crucial role in daily functional tasks such as walking and climbing stairs, as it controls the flexion and extension movements of the knee. In modern society, adults often spend prolonged periods of time sitting in various positions throughout the day. Therefore, to confirm the effect of these various posture characteristics on the quadriceps muscles, this study applied the three representative postures of sitting upright, sitting with legs crossed, and sitting with ankles crossed over knees for 15 min, and then during climbing and descending stairs, quadriceps muscle activity was confirmed. As a result, it was confirmed that sitting upright and sitting with ankles crossed over knees for 15 min had an immediate effect on the vastus medialis activity during descending the stairs. However, since this study confirmed the muscle activation pattern by applying only 15 min of sitting posture, it is not possible to clearly explain what effect the additional time has on quadriceps muscle activity, and because we investigated immediate differences in muscle activity, we do not understand how long-lasting changes in muscle activity are. Even, only healthy adults were targeted. Therefore, there is insufficient evidence that various postures for a long time can negatively affect knee dynamic stabilization in a group with knee joint disease. Thus, to better understand these effects, it is necessary to examine the difference in quadriceps muscle activity over time when sitting for 30 min or 1 h. In addition, it is emphasized that additional research is needed to confirm the change in quadriceps femoris activity before and after applying various sitting postures for a long-time targeting group with knee joint diseases such as anterior knee pain or patellofemoral pain syndrome.

Conclusions

In healthy women, there was no difference depending on the sitting posture, however prolonged sitting can lead to decreased muscle activity in the vastus medialis muscle, particularly during descending stairs in SAC. This reduction in muscle activity may negatively impact knee dynamic stabilization, which refers to the ability of the knee joint to maintain stability during movement. Therefore, the results of this study suggest the need to encourage dynamic relaxation in modern people. In future research, to understand the effect of sitting posture on quadriceps muscle activity, it is necessary to conduct research by recruiting groups with patellofemoral pain syndrome or knee pain, and understanding muscle activity over time after prolonged sitting may provide useful information in establishing criteria for determining functional recovery of the quadriceps femoris.

Supplemental Information

Supplemental Information 1 Codebook.

Supplemental Information 2 Raw data.

Supplemental Information 3 The raw data for SPSS statistical analysis.

This includes the results of individuals, not groups.

Supplemental Information 4 The statistical results produced based on the SPSS statistical analysis.

This includes normality tests, cross-tabulation, and inferential statistics on differences between groups.

Supplemental Information 5 STROBE Checklist.

The authors thank all the subjects who participated in this study.

Additional Information and Declarations

Competing Interests

Author Contributions

Human Ethics

Data Availability

The authors declare that they have no competing interests.

SaBin Chun conceived and designed the experiments, performed the experiments, analyzed the data, prepared figures and/or tables, authored or reviewed drafts of the article, and approved the final draft.

JongChul Park conceived and designed the experiments, analyzed the data, prepared figures and/or tables, and approved the final draft.

The following information was supplied relating to ethical approvals (i.e., approving body and any reference numbers):

Pukyong National University.

The following information was supplied regarding data availability:

The raw measurements are available in the Supplemental Files.

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
