# Peer review of "Exploring quadriceps activity variations in women: the role of seated posture during stair climbing"

_PeerJ, doi:10.7717/peerj.18547_

## Round 0.1 · original submission · Major Revisions

Dear Authors,

Please revise the manuscript considering the reviewers´ indications.

Thank you.

Best regards.

Reviewer 1 ·

Basic reporting

Although your introduction was quite clear, more details are required. To give more support for your study, I would suggest that you to improve the description at lines 75–80 (more specifically, you should elaborate on the knowledge gap being filled).

Experimental design

I have a few question related to your method
1. Irawan et al show similar method. Please explain why you used the opposite respondent criteria from the previous study. (Lines 95-96)
2. Based on your data, please explain your decision to use stratified random sampling based on IPAQ result

Validity of the findings

Interesting discussion, but you should add that the possible reason for changing VM activity only occurs in SU and SAC position during stair descending

Additional comments

I commend the authors for detailed fieldwork. In addition, the manuscript clearly written in professional structure. However needs some improvement in the method and discussion part (as I have noted above) which should be improved upon before Acceptance.

·

Basic reporting

Manuscript has been written with clarity. Language proficiency is simple and sceintific. Scientific Background can be updated with latest references (2019 onwards) if possible regarding various different postures how it affects quadriceps muscle fucntion. Raw data shared and it is giving originality to this manuscript. Hypothesis testing with statistical interfences has been expalined well.
Images are clear. All imges and tables were cited with legends properly.

For some minor revisions kindly go through annotated pdf file.

Experimental design

In methodology, Sample size calculation is missing and how the three groups of different sitting postures has been given should be mentioned clearly.
Otherwise the methodology part has been well expalined with intervention procedures andm outcomes meausrement.
This original research is clearly within the aim and scope of the journal.
Resaerch gap with research qusetion has been mentioned clearly with results and discussion section.
Methodology otherwise flawless.

Validity of the findings

Rationale for this study is clearly stated.
Discussion part can be elaborated why not much expected changes in vastus lateralis and rectus femoris muscle.
Conclusions are linjked to the research question and stated well to support the result findings of this study.
All datas have been provided.

Additional comments

Innovative appoarch to physiologically study the quadriceps muscle physiology. And this study may open new innovative way of prescribing exercises related to quadriceps muscle involvement in conditions like Osteoarthritis of knee joint.

Reviewer 3 ·

Basic reporting

Thank you for submitting your manuscript to the Journal of PeerJ. It is clear that the authors have done hard and heavy work to do and present this study. In general, the manuscript has novelty and good quality, but corrections are needed in some parts.

Basic Reporting Comments
1. The formatting and structure of the article require further attention and adherence to established guidelines. Please review the content again in accordance with the specified framework and reference an appropriate previously published article for alignment.
2. The selected title appears somewhat conventional. It may be prudent to consider alternative, more compelling titles that would better reflect the substantive value of your research. To that end, the following options are provided for your consideration:
o Impact of Seated Postures on Quadriceps Activation During Stair Climbing in Adult Women
o Evaluating Quadriceps Muscle Engagement in Different Seated Positions During Stair Ascension in Women
o The Influence of Sitting Posture on Quadriceps Performance During Staircase Ascent in Adult Females
o Exploring Quadriceps Activity Variations in Women: The Role of Seated Posture During Stair Climbing
o Assessment of Quadriceps Function in Adult Women: How Seated Position Affects Stair Climbing Efficiency
3. Based on my analysis of both my experience and the provided manuscript, it appears that a native speaker may not have authored the article. I recommend having it reviewed by a native speaker to identify and correct any spelling and grammatical errors. (e.g., climbing and descending stairs, ascending and descending from the stairs)
4. In the abstract section, please provide a concise summary of the statistical methods employed in the research within the methodology segment.
5. The conclusion section should be incorporated into the summary and articulated in a concise manner.
6. Please articulate some of the significant and meaningful findings from the research in the abstract results section.
7. I was unable to locate the keywords section in your manuscript. Please incorporate three keywords derived from the MeSH index into the manuscript.

Experimental design

Experimental Design Comments
8. Regarding the research sample size, could you please clarify the rationale behind selecting 36 participants? Additionally, why were the participants not evenly distributed with 12 individuals in each group?
9. I request a calculation of the appropriate sample size using established statistical formulas or G-Power software, incorporating parameters such as alpha, power, and a valid effect size based on the chosen statistical method. Furthermore, kindly attach the relevant documentation.
10. Could you please provide the rationale for each subject participating in the test three times? Would not the validity of the results be enhanced if each participant were to conduct the tests five times instead?
11. Please include the image associated with the staircase, detailing its dimensions, materials, and the precise number of steps taken by each participant.
12. Please specify, with clear references, the exact duration of rest intervals between each trial for all participants.
13. Please include photographs of the seating arrangement and the electrode placement pattern.
14. In what manner did the participants traverse the stairs: by stepping over each individual step or by ascending step by step?
15. Please provide the SPSS file for attachment.

Validity of the findings

Validity of Findings Comments
16. The articulation of the problem statement within the manuscript appears to lack the requisite clarity and cohesion. The substantive contents are described in a disconnected and fragmented manner, thereby failing to maintain proper thematic integrity.
17. A more rigorous and comprehensive problem formulation is warranted in order to establish a robust analytical foundation for the scholarly investigation. The problem statement should be structured in a manner that systematically delineates the key issues, challenges, and knowledge gaps that the research endeavor seeks to address.
18. The justification provided for the higher prevalence of the problem in women compared to men is insufficient. A more robust and scientifically grounded explanation is warranted to substantiate the observed gender differential.

Additional comments

General Improvement Comments
19. The introduction of the manuscript lacks thematic integrity and coherence. The paragraphs and sentences are not well-connected, and some of the background literature included is not directly relevant to the main topic. A revision of the introduction is needed to enhance its academic rigor and conceptual flow.
20. The manuscript's introduction does not effectively convey the importance, necessity, and novelty of the proposed research. A revision is needed to strengthen the justification for undertaking this scholarly investigation and to more comprehensively articulate its potential contributions.
21. The tables and figures presented are of insufficient quality and may lead to confusion for the readers. To enhance clarity and effectiveness, it is crucial to refer to articles published in this journal or other reputable sources for guidance on designing and formatting these visual elements. Improving the quality of the tables and figures can involve several steps:
o Clarity and Readability: Ensure that all text is legible and that the font size is appropriate. Avoid cluttering figures with excessive detail that can overwhelm the viewer.
o Consistent Formatting: Use consistent styles, colors, and symbols across all tables and figures to create a cohesive look. This helps readers easily navigate and compare information.
o Descriptive Labels: Provide clear, descriptive titles and labels for all axes and data points. This allows readers to quickly understand what each table or figure represents.
o Appropriate Types of Visuals: Choose the appropriate type of figure or table for the data being presented. For example, bar charts for comparisons or line graphs for trends can enhance comprehension.
o Referencing Standards: Familiarize yourself with the visual standards outlined in the journal’s guidelines or similar publications. This can help ensure that the figures and tables meet academic and publication standards.
22. In the conclusion section, the findings were simply restated without further elaboration. It is imperative that this section not only summarizes the results but also clearly articulates their practical implications. This involves discussing how the findings can be applied in real-world contexts, potential impacts on relevant stakeholders, and suggestions for future action or research based on the results. Additionally, incorporating specific recommendations or implications could enhance the conclusion's relevance and utility for practitioners in the field. By doing so, the conclusion can provide a more comprehensive understanding of the significance of the study's findings.
23. The discussion section of the article lacks coherence and consistency, making it difficult for readers to follow the narrative effectively. To improve this, it is essential to restructure the discussion to ensure a logical flow and clear connections between paragraphs. Here are some key steps to enhance the discussion:
o Establish a Clear Thesis: Begin the discussion with a strong statement that outlines the main findings and their significance. This will serve as a guiding framework for the rest of the section.
o Logical Segmentation: Organize the discussion into subsections that cover different aspects of the findings. Each subsection should focus on a specific theme or question, helping to maintain clarity.
o Smooth Transitions: Use transition sentences at the end of each paragraph to link ideas to the next. This will help create a smoother flow and guide readers through the reasoning.
o Cite Relevant Literature: Integrate references to existing literature throughout the discussion to provide context for the findings. This not only supports the arguments but also illustrates how they relate to prior research.
o Summarize Key Points: At the end of the discussion, briefly summarize the main points to reinforce the conclusions drawn from the research. This helps consolidate the information for the reader.
o Address Limitations and Future Directions: Clearly outline any limitations of the study and propose areas for future research. This adds depth to the discussion and encourages ongoing inquiry.

Annotated reviews are not available for download in order to protect the identity of reviewers who chose to remain anonymous.

---

## Round 0.2 · Minor Revisions

Dear Authors,

Please revise the manuscript considering the final minor suggestions provided by the reviewers.

Thank you.

Best regards.

Reviewer 1 ·

Basic reporting

The introduction section shows a clear research gap, However it can be clarified in the last paragraph (line 81–85)

Experimental design

The method section is written in detail with sufficient references and can be easily replicated

Validity of the findings

The conclusion section should include that there were no differences between sitting postures as mentioned in the results

·

Basic reporting

This manuscript seems to be with clarity about objectives of the study. Revised manuscript is good enough to be published.

Experimental design

There are no flaws in methodology. After revisions, the manuscript is clear.

Validity of the findings

On generalisability of this study findings need more evidence.

Reviewer 3 ·

Basic reporting

none

Experimental design

none

Validity of the findings

none

Additional comments

none

---

## Round 0.3 · accepted · Accept

Dear Dr. Chun,

Thank you for being so committed during the review process. Please pay special attention to details during the manuscript's proofing phase.

Best regards.

Reviewer 1 ·

Basic reporting

Clear purpose of study

Experimental design

experimental design is written clearly, easy to follow and replicable

Validity of the findings

the conclusion well stated and answer the research question